# Insight into the Molecular Signature of Skeletal Muscle Characterizing Lifelong Football Players

**DOI:** 10.3390/ijerph192315835

**Published:** 2022-11-28

**Authors:** Stefania Orrù, Esther Imperlini, Daniela Vitucci, Marianna Caterino, Annalisa Mandola, Morten Bredsgaard Randers, Jakob Friis Schmidt, Marie Hagman, Thomas Rostgaard Andersen, Peter Krustrup, Margherita Ruoppolo, Pasqualina Buono, Annamaria Mancini

**Affiliations:** 1Department of Movement Sciences and Wellness, University Parthenope, 80133 Naples, Italy; 2CEINGE-Biotecnologie Avanzate Franco Salvatore, 80145 Naples, Italy; 3Department for Innovation in Biological, Agro-Food and Forest Systems, University of Tuscia, 01100 Viterbo, Italy; 4Department of Molecular Medicine and Medical Biotechnology, School of Medicine, University of Naples Federico II, 80131 Naples, Italy; 5Department of Sports Science and Clinical Biomechanics, University of Southern Denmark, 5230 Odense, Denmark; 6Section for Anaesthesia for ENT, Head Neck & Maxillofacial Surgery and Ortopedi, Rigshospitalet, Copenhagen University Hospital, 2100 Copenhagen, Denmark; 7Sport and Health Sciences, College of Life and Environmental Sciences, St. Luke’s Campus, University of Exeter, Exeter EX1 2LU, UK; 8Danish Institute for Advanced Study (DIAS), University of Southern Denmark, 5230 Odense, Denmark

**Keywords:** lifelong football training, successful aging, mitochondrial oxidative capacity, polyamines

## Abstract

Background: Aging and sedentary behavior are independent risk factors for non-communicable diseases. An active lifestyle and structured physical activity are positively associated with a healthier quality of life in the elderly. Here, we explored the proteomic/metabolomic muscular signature induced by lifelong football training associated with successful aging. Methods: The study was performed on nine lifelong football players (67.3 ± 2.8 yrs) and nine aged-matched untrained subjects. We performed a proteomic/metabolomic approach on *V. lateralis* muscle biopsies; the obtained data were analyzed by means of different bioinformatic tools. Results: Our results indicated that lifelong football training is able to enhance the muscles’ oxidative capacity in the elderly by promoting fatty acids as preferential energetic substrates and hence determining a healthier body composition and metabolic profile; furthermore, we showed that the total polyamine content is higher in lifelong football players’ muscle, enforcing the involvement of polyamines in muscle growth and hypertrophy. Conclusions: Lifelong football training, as a structured physical activity, significantly influences the expression of the proteins and metabolites involved in oxidative metabolism and muscle hypertrophy associated with successful aging.

## 1. Introduction

Aging is a continuous physiological process of the human body that begins in early adulthood with the onset of multiple changes at the cell/tissue level (cellular homeostasis alteration, DNA accumulation and protein damage) [1,2,3], eventually leading to a functional decline in organs/systems [4,5,6]. In fact, this structural and functional decay affects the cardiovascular, the musculoskeletal and the nervous system, initiating various age-related non-communicable diseases (NCDs), such as Type 2 diabetes mellitus (T2DM), hypertension, insulin resistance (IR), cognitive decline and Alzheimer’s disease [7,8]. Another independent risk factor for major NCDs is extensive sedentary behavior (SB), which is defined as any waking behavior characterized by an energy expenditure of 1.5 METs (metabolic equivalent of task, corresponding to the amount of oxygen consumed at rest) or less while sitting, lying, or leaning [9,10,11]. Many older adults are less active than recommended [12]. However, SB does not mean “physical inactivity”. In fact, older people can meet the current recommendations for moderate to vigorous physical activity (MVPA) but be sedentary in other daily activities; on the contrary, they could have a low SB while not being engaged in any structured PA [10,13]. NCDs due to aging and SB can be counteracted through an active lifestyle and planned PA, which are positively associated with a better quality of life in the elderly. Several studies have reported that individuals who engage in PA age successfully, characterized by a low risk of age-related disabilities in comparison with age-matched sedentary adults [14,15]. Regular physical activity, in fact, exerts its effects on metabolic regulation and the maintenance of metabolic homeostasis [16], especially when the WHO’s guidelines, which recommend both aerobic and resistance exercise, are met [3].

In this context, football training represents an intermittent exercise that includes millions of active players of different ages, genders and pathophysiological conditions around the world [17]. Football training, in fact, is characterized by high intensity anaerobic actions interspersed with periods of low-intensity recovery [18]. For this reason, it is an effective tool to stimulate physiological and molecular adaptations leading to an improved health status at all ages, from young to middle-aged and older people [19,20,21,22]. As demonstrated, long-term recreational football training improves cardiovascular, skeletal muscle and metabolic fitness, and health-related body composition parameters. Moreover, it increases the expression of key proteins involved in oxidative metabolism, mitochondrial biogenesis, DNA repair, autophagy and protein quality control through the regulation of specific miRNAs, such as miR-1303 [23], together with heat shock proteins (HSP), such as HSP 70/90 and the components of the proteasome protein complex [21,24,25,26]. Although several molecular markers involved in the adaptation of skeletal muscle to PA and in protective mechanisms against age-related NCDs have been unveiled over the past decade, many questions have yet to be resolved. Here, we explored the different muscular signatures from veteran soccer players (VPG) vs. age-matched active untrained healthy subjects (control group, CG) at the molecular level. We reported the results of a proteomic/metabolomic approach performed on *V. lateralis* muscle biopsies in order to highlight the molecular mechanisms that characterize successful aging through lifelong football training.

## 2. Materials and Methods

### 2.1. Subjects

The study was performed on 18 healthy male volunteers aged 64–71 years, who enrolled at the University of Copenhagen. Nine were football players who, in the last 10 years, had trained for one session per week (1.5 ± 0.6 h/session) and played in 26 ± 12 football matches (2 × 35 min) per year in local football clubs in Copenhagen; they constituted the veteran football player group (VPG; 67.3 ± 2.8 yrs). The control group (CG; 66.5 ± 1.6 yrs) were nine healthy aged-matched active untrained men. For both groups, the exclusion criteria were a history or symptoms of cardiovascular disease or cancer, Type 2 diabetes, hypertension, nephropathy or musculoskeletal complaints, whereas the inclusion criterion was the ability to perform all the tests required for study participation as previously described [21,25,26]. The anthropometric, biochemical and clinical parameters of the participants are shown in Table 1 and were evaluated as previously reported [21,25,26]. Moreover, the body composition, maximal oxygen uptake (VO_2_max) and resting heart rate (RHR; bpm) were measured for all subjects as described by Mancini and colleagues [21,25,26]. All subjects were informed about any potential discomforts or risks related to the experimental protocol and gave their informed written consent to participate in the study. This was conducted according to the Declaration of Helsinki and was approved by the local ethics committee of the University of Copenhagen (H-1-2011-013, ClinicalTrials.gov Identifier: NCT01530035).

### 2.2. Muscle Sample Collection and Preparation

Before muscle biopsy, all recruited subjects observed an overnight fast. Muscle biopsies were obtained from the vastus lateralis under local anesthesia by using the Bergstrom technique [21]. The muscle samples were immediately frozen in liquid nitrogen and stored at −80 °C until subsequent analysis. Muscle biopsies were homogenized in a buffer containing 150 mM NaCl, 1% Triton, 5 mM EDTA, 50 mM Tris-HCl and the Complete Mini protease inhibitory cocktail. After 30 min of incubation on ice, the samples were clarified by centrifugation at 16,000× *g* for 30 min at 4 °C. The protein concentration was determined using Bradford’s reagent (Bio-Rad, Hercules, CA, USA).

### 2.3. Electrophoretic Separation and In-Gel Digestion

For each group (VPG and CG), the nine protein samples were pooled, incubated for 5 min at 95 °C and separated by 10% SDS-PAGE. Gel bands were stained with Colloid Blue Stain Reagent (Thermo Fisher Scientific, Waltham, MA, USA) according to the manufacturer’s procedure. Gel images were acquired using the scanner GS-800 Calibrated Densitometer (Bio-Rad, Hercules, CA, USA). The whole gel lanes were manually cut into 2 mm gel slices and further processed as previously described [27]. In particular, each slice was first washed with acetonitrile (ACN) and then with 50 mM ammonium bicarbonate. Protein bands were reduced by incubation with 10 mM DTT for 45 min at 56 °C and alkylated in 55 mM iodoacetamide for 30 min in the dark at room temperature. Enzymatic digestion was carried out with 10 ng/μL of modified trypsin (Promega, Madison, MI, USA) in 50 mM ammonium bicarbonate for 45 min at 4 °C. Peptide mixtures were extracted as previously reported and resuspended in 0.2% (*v*/*v*) formic acid for mass spectrometry (MS) analysis.

### 2.4. LC-MS/MS Analysis, and Protein Identification and Quantitation

Mass spectrometry analysis was performed by using an LTQ-Orbitrap XL (Thermo Fisher Scientific, Bremen, Germany) equipped with the nanoEASY II Nanoseparations chromatographic system (75 μm–L, 20 cm column, Thermo Scientific, Bremen, Germany) as previously described [28]. The peptide analysis was performed at a resolution of 30,000 and used the data-dependent acquisition of an MS scan (400–1800 m/z) followed by MS/MS scans of the five most abundant ions. Raw MS/MS data in Mascot format text (mgf) were processed by the Proteome Discoverer platform (version 1.4, Thermo Fisher Scientific, Waltham, MA, USA) interfaced with an in-house Mascot server (version 2.3, Matrix Science, London, UK). Proteins were identified with the following search parameters: UniProt as the database, limited to the *Homo sapiens* taxonomy; trypsin as a specific proteolytic enzyme; 1 missed cleavage allowed; 10 ppm precursor tolerance and a 0.6 Da fragment ion tolerance; carbamidomethylation of cysteine as a fixed modification; conversion of N-terminal glutamine to pyro-glutammic acid and oxidation of methionine as variable modifications. Only proteins with at least 2 assigned peptides with an individual Mascot score > 19 were considered to be significant. For the label-free quantitative analysis, spectral counts (SpC) values were used to estimate the proteins’ abundance, as previously reported [29]. 

In order to perform a semi-quantitative comparative analysis, the protein abundances in each of the proteomes considered were expressed as Rsc, calculated according to the following formula:Rsc = log2 [(n2 + f)/(n1 + f)] + log2 [(t1 − n1 + f)/(t2 − n2 + f)]

Specifically, Rsc is the log ratio of abundance between Samples 1 (VPG) and 2 (CG); n1 and n2 are the SpCs for the given protein in Groups 1 and 2, respectively; t1 and t2 are the total number of spectra of all proteins in the two sample groups; and f is a correction factor set to 0.5 that was used to eliminate discontinuity due to SpC = 0 [30].

In this study, proteins with Rsc ≥ 1.40 or ≤ −1.40 were considered to be differentially represented in VPG versus CG groups. The mass spectrometry proteomics data have been deposited at the ProteomeXchange Consortium (available at http://www.proteomexchange.org, accessed on 28 October 2022) via the PRIDE [31] partner repository with the dataset identifier PXD037792.

### 2.5. Bioinformatic Analysis

The differentially expressed proteins identified here were classified according to the Database for Annotation, Visualization and Integrated Discovery (DAVID) (version 6.7, http://david.abcc.ncifcrf.gov/, accessed on 11 November 2020). Based on Fisher’s exact test, the DAVID tool determines the protein enrichment in the Gene Ontology (GO) annotation terms, particularly biological processes. Only annotation categories with a *p*-value of ≤0.05 were considered to be significant. Moreover, the Search Tool for the Retrieval of Interacting Genes (STRING) (version 11, http://string-db.org/, accessed on 11 November 2020) was used for generating the protein–protein interaction networks. The STRING tool imports known and predicted protein–protein associations including physical and indirect interactions. An interaction score of 0.7 (high confidence) was set, considering the following sources: neighborhood, co-occurrence, co-expression, experiments and databases. STRING was also used to analyze the most statistically significant and non-redundant biological processes among the differentially expressed proteins from the proteomic dataset. Only annotation biological processes with a false discovery rate (FDR) ≤ 0.05 were considered significant. In addition to the identification of functional annotations and biological networks, a pathway analysis was also performed according to the Reactome database (https://www.reactome.org, accessed on 11 November 2020).

### 2.6. Western Blotting Analysis

Protein extracts (30 µg) from three independent replicates of the VPG and CG were resolved on a 4–20% precast gradient of polyacrylamide gels (Bio-Rad) and then transferred onto nitrocellulose membranes by using the Bio-Rad Trans-Blot Turbo apparatus. The membranes were analyzed by Western blotting as previously described [25] and incubated with the following primary antibodies: rabbit anti-CPT1B (1:1000, Abcam, Cambridge, UK) and mouse anti-ACAA2 (1:500, Santa Cruz, Dallas, TK, USA). A mouse anti-GAPDH (Sigma-Aldrich, St. Louis, MI, USA) antibody was used as a loading control at a dilution of 1:500. Immunoblot detection was carried out using horseradish peroxidase-conjugated anti-mouse (1:5000) or anti-rabbit (1:5000) secondary antibodies (GE Healthcare, Chicago, IL, USA) and the enhanced chemiluminescence advanced Western blotting detection kit (GE Healthcare). The signals were visualized by X-ray film exposure. Digital images were acquired by GS-800 calibrated densitometer scanning (Bio-Rad). Densitometric measurements were made with Image J2 software; in particular, the raw densitometry signal of each protein band was quantified and normalized against the corresponding GAPDH band. The results were shown as the percentage of the mean of controls and statistically evaluated by Student’s *t*-test (*p*-values < 0.05).

### 2.7. MS-Based Metabolite Profiling

For metabolite extraction, muscle biopsies from the VPG and CG were homogenized in 1 mL of 50:50 cold methanol/0.1 M HCl and then centrifuged at 15,000× *g* for 30 min at 4 °C to recover the supernatant containing the metabolites; whereas proteins were extracted from the pellet to estimate the protein concentration as previously described [32]. The metabolite mixture was dried under nitrogen and analyzed by MS/MS to evaluate the amino acid (AA) and acylcarnitine (AC) levels. The metabolite sample was resuspended in methanol containing standard mixtures of labeled AA and AC, incubated for 20 min at room temperature and dried under nitrogen [33]. The metabolites were esterified with 3N HCl/n-butanol at 65 °C for 25 min. The derivatized samples were dried again under nitrogen and resuspended in 300 μL of acetonitrile/water (70:30) containing 0.1% formic acid. For the MS/MS analyses, 100 μL of three independent aliquots of each sample was injected in the API 4000 triple quadrupole mass spectrometer (Applied Biosystems-Sciex, Toronto, ON, Canada) coupled with an 1100 series Agilent high-performance liquid chromatography system (Agilent Technologies, Waldbronn, Germany). MS/MS analyses for AAs and ACs were performed according to parameters that have been previously described [33]. The MS/MS data were quantitatively analyzed by comparing the metabolites and the corresponding internal standard areas using ChemoView v1.2 software. The AA and AC contents were normalized against the total amount of protein from the cellular extract. MetaboAnalyst 4.0 (http://www.metaboanalyst.ca, accessed on 15 January 2021) was used to perform the multivariate statistical analysis. The metabolome dataset, including 50 metabolites’ concentrations, was log10-transformed and scaled according to the Pareto scaling method. Partial least squares-discrimination (PLS-DA) was carried out to evaluate the dataset’s homogeneity and estimate the variable importance in projection (VIP). VIP is the weighted sum of squares of the PLS loadings, taking the amount of variation explained in each dimension into account. Univariate statistical analysis was carried out using GraphPad Prism 9.0, and the results are presented as the mean ± standard error of the mean (SEM). The statistical significance of the difference in the metabolite samples’ concentrations between two different groups was evaluated by parametric (unpaired *t*-tests with Welch correction) or non-parametric (Mann–Whitney tests) tests. The normal distribution was verified according to D’Agostino and Pearson’s test.

### 2.8. Polyamine Assay

The total polyamine content in muscle biopsies from the VPG and CG was measured by using a fluorimetric assay kit (Abcam, Cambridge, UK), according to the manufacturer’s protocol. Briefly, muscle biopsies were homogenized on ice using a Dounce homogenizer and centrifuged at 10,000× *g* for 5 min at 4 °C to collect the supernatants containing the metabolites. The samples were treated with a sample clean-up reagent and then filtered through a 10 kDa spin column to avoid common metabolites other than polyamines interfering with the assay. The assay kit includes a selective enzyme mix that acts on polyamines to generate hydrogen peroxide and includes a specific fluorometric probe to detect a fluorescence signal proportional to the amount of polyamine after the reaction of the hydrogen peroxide produced. According to the manufacturer’s protocol, the samples were prepared in a black 96-well plate and the fluorescence was read using a Perkin Elmer Enspire plate reader (Perkin Elmer, Waltham, MA, USA) at 535 and 587 nm as the excitation and emission wavelengths, respectively. The total polyamine content was quantified using a calibration curve established with known amounts of polyamine standards. The experiments were performed using five independent biological replicates, each with three technical repeats.

## 3. Results

### 3.1. Identification of Differentially Expressed Proteins in the Skeletal Muscle from Veteran Football Players (VPG) versus Untrained Subjects (CG)

To investigate whether lifelong football training affects the protein expression profiles in the skeletal muscles of the lower limb, a label-free differential proteomic study was carried out to analyze muscle biopsies from the *V. lateralis* of 18 healthy male volunteers, nine belonging to the veteran football player group (VPG) and nine healthy age-matched untrained subjects (control group, CG), enrolled at the University of Copenhagen. The two groups showed no differences in their anthropometric, biochemical and clinical parameters, but the BMI (24.7 ± 1.7 kg/m^2^ vs. 29.6 ± 4.3 kg/m^2^, *p* < 0.05) and body fat percentage (22.9 ± 6.5% vs. 33.4 ± 5.0%, *p* < 0.05) were significantly lower and the VO_2_max (34.8 ± 1.5 mL/min/kg vs. 25.2 ± 3.1 mL/min/kg, *p* < 0.001) was significantly higher in the VPG than in the CG (Table 1). After extraction, muscle protein mixtures from the VPG were pooled and separated by monodimensional SDS-PAGE; the same procedure was applied to muscle protein mixtures from the CG (Appendix A). Coomassie-stained protein bands were excised, in-gel digested and analyzed by LC–MS/MS, and their raw data were further analyzed by the Proteome Discoverer platform to allow identification and quantification of the proteins. We identified 873 unique proteins, which are listed in Appendix A. To define a map of the differentially expressed species between the VPG and CG, a label-free quantitative analysis was performed based on the MS spectral counts of the identified proteins. We found 188 differentially expressed proteins in the VPG versus CG; Table 2 and Table 3 list the 92 overexpressed and 96 underexpressed species, respectively. For each protein, the Uniprot accession code, the description, the gene name and the Rsc value are reported.

### 3.2. Functional Annotation, Biological Network and Pathway Analyses of Differentially Expressed Proteins in Skeletal Muscle from VPG versus CG

To investigate the molecular basis related to healthy longevity in the skeletal muscle from the VPG versus CG, under- and overexpressed protein datasets were analyzed separately by bioinformatic tools. Table 4 lists the most significant Gene Ontology (GO) terms belonging to the BP (biological processes) category related to the two subsets obtained by querying the DAVID platform. Among the overexpressed proteins in skeletal muscle from the VPG versus the CG, the most represented functional categories were “mitotic cell cycle”, “calcium ion transport” and “energy derivation by oxidation of organic compounds”; whereas “protein complex assembly”, “positive regulation of ubiquitin–protein ligase activity” and “organic acid catabolic process” were all categories that were significantly enriched in the underexpressed protein subset. Differentially expressed proteins were also analyzed by STRING software version 11.5 to unveil the relevant biological networks and non-redundant biological processes in both protein subsets. The STRING output revealed three and five relevant subnetworks among the over- and underexpressed proteins in the VPG vs. CG, respectively (Figure 1). In agreement with the DAVID analysis (Table 4), the STRING subnetworks of overexpressed proteins contained nodes belonging to the following biological processes: “nuclear-transcribed mRNA catabolic process”, “regulation of calcium ion transport” and “oxidation–reduction process” (Table 5, Figure 1A). Similar to the DAVID analysis, the underexpressed protein nodes in the STRING analysis were also related to “protein-containing complex subunit organization” and “post-translational protein modifications” (Table 5, Figure 1B). Interestingly, 26S proteasome subunits (PSMB4, PSMC5, PSMB1, PSMC4), which were underexpressed proteins in the VPG vs. CG, were significantly represented in the BP term “positive regulation of ubiquitin-protein ligase activity” according to DAVID (Table 4) and highly interconnected in the STRING subnetwork related to “post-translational protein modifications” (Figure 1B). When the Reactome database was queried, these four 26S proteasome subunits matched the “regulation of ornithine decarboxylase (ODC)” pathway (*p*-value = 1.13 × 10^−10^, FDR 5.40 × 10^−9^), a cellular process affecting the endogenous biosynthesis of polyamines. 

### 3.3. Validation of Selected Differentially Expressed Proteins

To validate the data obtained from the label-free quantitative proteomics, we analyzed the expression of selected identified proteins by Western blotting (Figure 2). In particular, we focused on two proteins, carnitine O-palmitoyltransferase 1, muscle isoform (CPT1B), and mitochondrial 3-ketoacyl-CoA thiolase (ACAA2), both belonging to the fatty acid degradation pathway. In agreement with the proteomic analysis, Western blotting showed the overexpression of these proteins in skeletal muscle from the VPG vs. CG (Figure 2A). This finding was confirmed by a densitometric analysis that showed, in the case of the overexpression of CPT1B and ACAA2, a significant difference between the VPG and CG (*p* < 0.05).

### 3.4. MS-Based Profiling of Free Amino Acids and Acylcarnitines in Skeletal Muscle from Veteran Football Players (VPG) versus Untrained Subjects (CG)

A targeted metabolomic analysis was performed to characterize the muscle biopsies from the VPG and CG (Figure 3). A detailed list of the identified metabolites, including their names, abbreviations and analytical concentrations is reported in Appendix A.

To define the VPG’s specific metabolomic signature, the datasets were processed according to a supervised partial least squares-discriminant analysis (PLS-DA). The VPG’s and CG’s metabolomes clustered according to variance of Component 1 (15.4%) and Component 2 (6.9%) (Figure 3A). The most discriminant hits between the VPG and CG were defined by evaluating the variable importance in projection (VIP) (Figure 3B).

In detail, the relative abundance of metabolites such as C16:1OH, C16:1, C14OH, C16, C18:1OH, C14:2 and Orn were able to discriminate between the analyzed groups according to a VIP > 1.5. Interestingly, the Euclidean distance-based hierarchical clustering of the identified and quantified metabolites visualized in the heatmap (Figure 3C) exhibited a clear distinct pattern in the metabolites’ abundance between the two groups, VPG and CG, with the exception of VPG1. The metabolite concentrations were ranked by their t-test (*p* < 0.05) results.

In order to highlight individual metabolite profiles between the VPG and CG, a univariate binary comparison was performed according to a volcano plot. Figure 3D summarizes the significant differences in the abundance of muscle metabolites between the two groups. The volcano plot analysis revealed that 7/50 and 9/50 metabolites increased and decreased, respectively, in the muscle samples from the VPG and CG.

Finally, the normal distribution of the metabolite concentrations was verified, and significant differentially profiles were evaluated in the different groups. In detail, metabolites such as Orn, Phe, Gly, Met, Arg were found to be increased in the VPG vs. CG; conversely, C16, C5DC, C14OH, C16:1, C8dc, Cit, C12Oh, C14:2 and C18 were found to be decreased in the VPG vs. CG (Figure 3E).

### 3.5. Polyamine Biosynthesis in Skeletal Muscle from Veteran Football Players (VPG) versus Untrained Subjects (CG)

The Orn/Cit profiles determined by a metabolomic approach, and the bioinformatic output obtained from the Reactome database suggested that we should verify the total polyamine content in the muscle biopsies from the VPG and CG by using a fluorimetric assay. As shown in Figure 4, the total polyamine content in the muscle was 39% higher in the VPG in comparison with the CG (*p* < 0.05).

## 4. Discussion

In-depth knowledge of the molecular mechanisms that govern the aging process in the skeletal muscle are instrumental to better counteract the structural and functional decay that occurs in the elderly [34,35]. Several studies have analyzed, at the molecular level, the effects of PA on different systems/organs, including the skeletal muscle, in the young, adults and older people [22,23,25,26,35,36]. Here, we characterized muscle biopsies from the *V. lateralis* of male subjects aged over 65 yrs by a multi-omics approach in order to gain an insight into the molecular mechanisms characterizing successful aging through lifelong football training. The anthropometric, biochemical and clinical data of the two groups under investigation differed in their VO_2_max, which was significantly higher in the VPG, and in the BMI and body fat percentage, which were significantly lower in the VPG. The proteomic approach identified 188 differentially expressed proteins between the two groups, 40 of which were from the mitochondria. Mitochondrial species were mostly upregulated (75%), similar to the results of previous reports [35,36]; among them, half were from the mitochondrial membrane. These proteins covered a variety of mitochondrial functions, as already described for high-functioning octogenarian master athletes [35], and supported the established knowledge that regular PA improves mitochondrial function [37,38,39,40,41], counteracting the decline in muscle strength and mass and in neuromuscular control [42,43]. In fact, the exercise-mediated coordinated expression of mitochondrial species activates mitochondrial biogenesis, which affects several functions within the organelle and triggers several signaling pathways [44,45] that are able to induce specific adaptations to the mechanical stimuli; among these, the increased oxidative capacity is a well-recognized hallmark of PA-mediated adaptation in the mitochondria [41,46,47]. Accordingly, by using different bioinformatic tools, in the overexpressed subset of this study, mitochondrial proteins involved in the oxidative metabolism were found, such as NDUFA7, NDUFA9, SUCLG1, UQCRH, ACAA2, ACADSB and ATP5J, confirming the previous findings. Moreover, the increase in the cytosolic Ca^2+^ concentration determined the overexpression of PGC-1α [25] and of the beta, delta and gamma subunits of CaMK2, a Ca^2+^/calmodulin-dependent protein kinase, which is involved in contraction-induced signaling [46], which favors lipid oxidation and supports skeletal muscles in using FAs as energetic substrates [45]. In agreement with such findings, in our dataset, we also found the overexpression of CPT1B, a rate-limiting enzyme in lipid oxidation, and the underexpression of the mitochondrial acetyl-CoA synthetase 2 (ACSS1), an enzyme involved in FA biosynthesis. A similar difference in expression was reported by Joseph et al. [48] in the skeletal muscle of rats following exercise, demonstrating the crucial role played by CaMK2 in enhancing the oxidative capacity of the mitochondria by both promoting the expression of enzymes involved in the oxidation of FAs and reducing the expression of those that catalyze lipid synthesis [48]. All together, these data indicate that the rate of FA beta oxidation was very high in the VPG compared with the CG, caused by the increased amount of the acyltransferase CPT1B, which promotes the entry of long-chain FAs into the mitochondria, and of ACADSB, ACAD8 dehydrogenases and ACAA2 thiolase, which are directly involved in the mitochondrial oxidative process. The proteomic picture was also supported by the metabolomic analysis, in which unsaturated, branched and hydroxylated ACs, considered to be intermediate metabolites of FA beta oxidation, were significantly reduced in the VPG compared with the CG. As a matter of fact, the increased mitochondrial oxidative capacity in the VPG resulted in the rapid disposal of lipidic intermediates due to the more efficient FA beta oxidation, whereas the lack of any structured PA in the CG contributed to the accumulation of FA metabolites. Such results confirmed that lifelong football training is able to enhance the muscles’ oxidative capacity in the elderly [25] by promoting lipids as preferential energetic substrates and hence determining a healthier body composition and metabolic profile.

Another interesting result obtained from our multi-omics approach is related to the regulation of ODC pathway, which emerged as significant process in the bioinformatic analysis comparing the proteomic data of the VPG and CG. ODC catalyzes the first rate-limiting step of putrescine synthesis, the first member of the polyamine family, under the stimulus of several growth factors, including exercise-sensitive circulating species. Putrescine, synthesized from the amino acid ornithine, is converted by the S-adenosylmethionine decarboxylase in spermidine and then in spermine [49,50,51].

Polyamines are small aliphatic polycations, the concentration of which is tightly regulated in their biosynthesis, catabolism and transport. They are involved in cell growth, proliferation and differentiation, and also in aging, metabolic diseases, cancer and neurodegenerative disorders; they also have the function of stabilizing the DNA and modulating some membrane receptor complexes [49,50,52].

Polyamines play several roles in the cell, aimed at protecting against oxidative stress by regulating the expression of proteins involved in the response to an oxidizing agent/condition [53,54,55,56,57,58]; they also act as ROS scavengers at a physiological pH [59,60,61], with spermine being the most effective at protecting DNA from oxidative stress [59]. Information on the role of exercise-induced polyamines in the skeletal muscle is still scarce. According to Turchanowa et al., polyamines are apparently involved in the oxidative metabolism of skeletal muscles, as their concentrations increased after resistance and/or endurance exercises [52,62]. Interestingly, in our experimental dataset, metabolomic data on the amino acids showed that in the VPG, ornithine concentrations were higher and, concurrently, citrulline concentrations were lower in respect to the CG’s muscle biopsies, supporting the hypothesis that ornithine is committed to polyamine biosynthesis through ODC’s activity. The fluorimetric assay confirmed that the total polyamine content in skeletal muscles from the VPG was higher than in the CG. Such a finding reinforces the putative involvement of polyamines in muscle growth and hypertrophy [49], and it encourages further investigation into the role of these exercise-sensitive metabolites.

## 5. Conclusions

This multi-omics study characterized the protein and metabolic profile from muscle biopsies of the VPG vs. CG. The study showed that mitochondrial biogenesis is effectively triggered in subjects who practice lifelong football training by means of the activation of CAMKII signaling and a very efficient mitochondrial oxidative process of FA; in addition, the concentration of small molecules, such as polyamines, which are known to be ROS scavengers and to have anti-inflammatory properties, was increased in the veterans compared with the controls. These results suggest that lifelong football training, as structured PA, significantly influences the expression of proteins and metabolites involved in successful aging and help reduce the risk of the onset of NCDs in the elderly.

## Figures and Tables

**Figure 1 ijerph-19-15835-f001:**
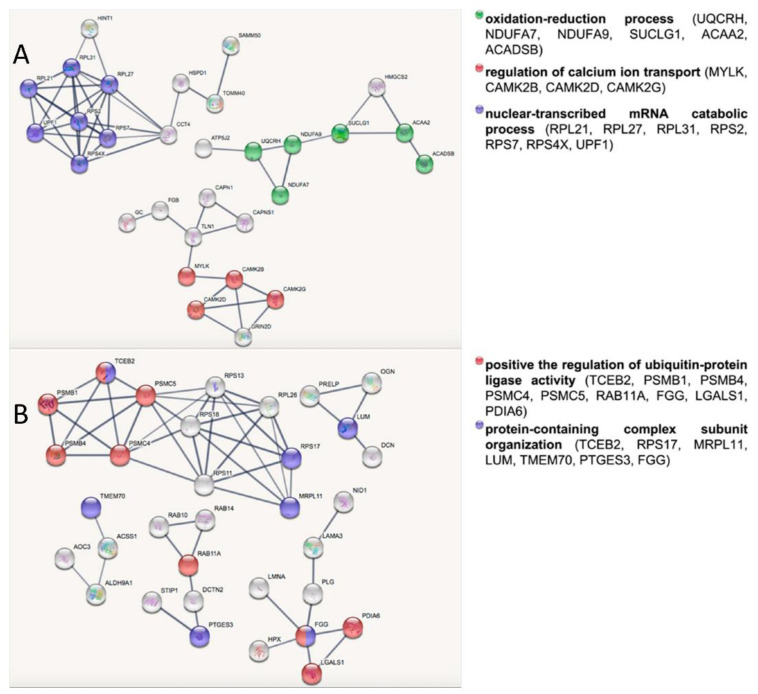
STRING network analysis of overexpressed (**A**) and underexpressed (**B**) proteins in skeletal muscle from the VPG vs. CG. The node colors correspond to the enriched categories with a FDR ≤ 0.05.

**Figure 2 ijerph-19-15835-f002:**
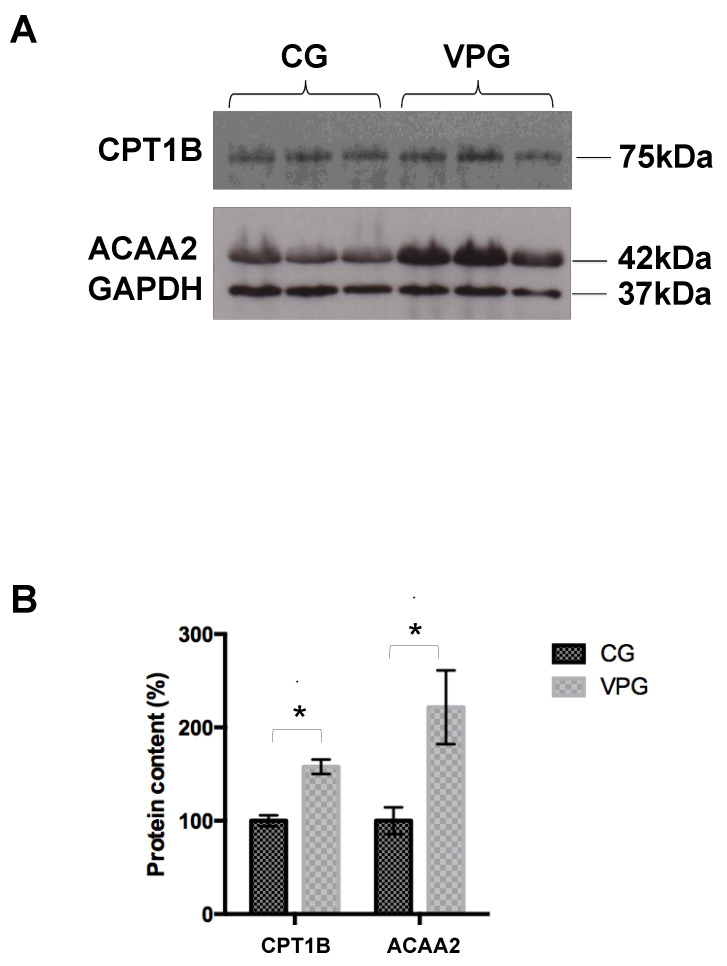
(**A**) Western blotting analysis of selected differentially expressed proteins in skeletal muscle from the VPG vs. CG. (**B**) Densitometric analysis of CPT1B and ACAA2. Values were normalized against GAPDH. The results, expressed as percentages, are shown as means ± SD. * *p* < 0.05.

**Figure 3 ijerph-19-15835-f003:**
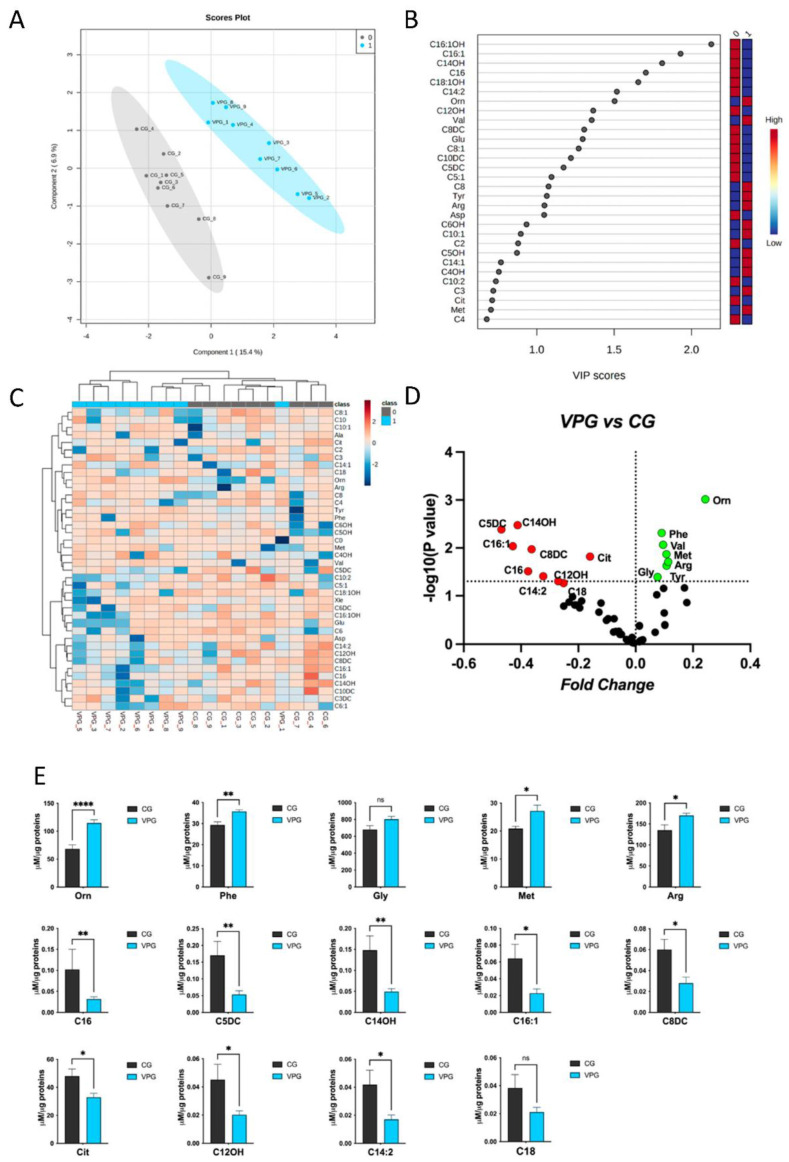
Metabolomic discriminant analysis of the VPG. (**A**) A supervised partial least squares-discriminant analysis (PLS-DA) was performed using the metabolome dataset. Muscle metabolite concentrations were log(10)-transformed and Pareto-scaled. A clear separation according to Component 1 (15.4%) and Component 2 (6.9%) was obtained. (**B**) The 30 most discriminant features were identified according to the variable importance in projection (VIP) score (>1.0). (**C**) Hierarchical cluster analysis and heatmap visualization of the top 40 members of the lipid dataset (y-axis), ranked by the *t*-test (*p* < 0.05) results are shown. (**D**) Volcano plot analysis, showing the metabolites’ abundance in the VPG versus CG. The relative abundance of each metabolite was plotted against its statistical significance as the fold change (log2 ratio) and −log10 (*p*-value). Red and green dots indicate significantly decreased and increased metabolites. Black dots indicate the metabolites identified in the dataset for which the relative abundance was not significantly changed between the VPG and CG. (**E**) Plots showing the different metabolite concentrations (means ± SEM) in the VPG versus CG. The significant difference in the metabolite concentrations was evaluated by performing a parametric *t*-test with Welch’s correction in normally distributed datasets and the Mann–Whitney test in non-normally distributed datasets (* *p* < 0.05, ** *p* < 0.01, **** *p* < 0.0001, ns = not significant). The normal distribution was verified according to D’Agostino and Pearson’s test.

**Figure 4 ijerph-19-15835-f004:**
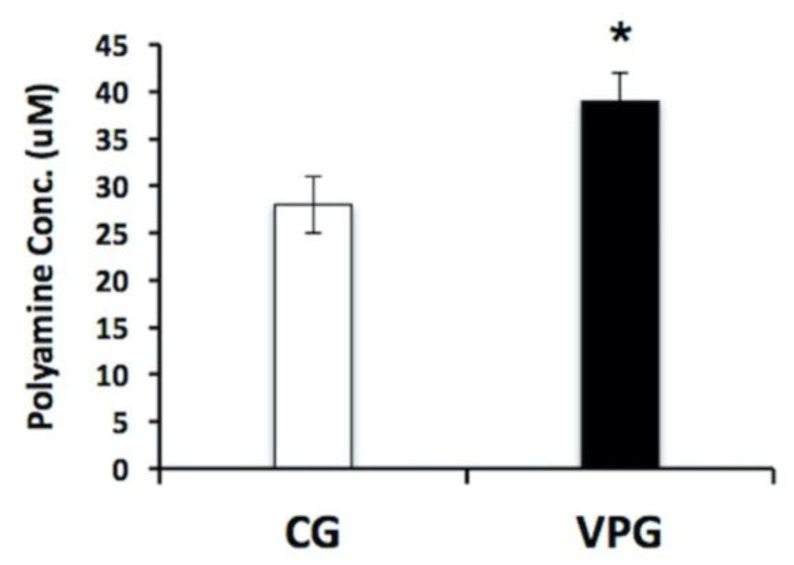
Determination of total polyamine content in skeletal muscle from the VPG vs. CG using a fluorimetric assay. Fluorescence signals proportional to the polyamine concentration in the muscle biopsies were obtained from three technical measures in five independent biological replicates. Results are represented as the mean ± SD. * *p* < 0.05.

**Table 1 ijerph-19-15835-t001:** Anthropometric, biochemical and clinical parameters of the subjects participating in the study.

	VPG	CG
**Number of subjects**	9	9
**Age (yrs)**	67.3 ± 2.8	66.5 ± 1.6
**Height (cm)**	179.2 ± 2.2	175.0 ± 5.2
**Body weight (kg)**	79.3 ± 6.4	92.7 ± 14.6
**BMI (kg/m^2^)**	24.7 ± 1.7 *	29.6 ± 4.3
**Fat mass (%)**	22.9 ± 6.5 *	33.4 ± 5.0
**Lean mass (kg)**	57.3 ± 1.9	57.9 ± 7.6
**VO_2_max (mL/min/kg)**	34.8 ± 1.5 **	25.2 ± 3.1
**Fasting blood glucose (mmol/L)**	5.3 ± 0.4	5.5 ± 0.5
**Total cholesterol (mmol/L)**	5.4 ± 1.4	5.4 ± 0.5
**HDL cholesterol (mmol/L)**	1.6 ± 0.8	1.2 ± 0.1
**LDL cholesterol (mmol/L)**	3.1 ± 1.2	3.3 ± 0.5
**Triglycerides (mmol/L)**	0.9 ± 0.5	1.1 ± 0.4
**Systolic BP (mmHg)**	124.7 ± 16.4	131.2 ± 12.9
**Diastolic BP (mmHg)**	70.2 ± 4.2	74.7 ± 10.1
**Mean arterial pressure (mmHg)**	88.3 ± 7.7	93.5 ± 10.7
**Resting heart rate (bpm)**	49.2 ± 5.9	61.0 ± 14.5

Values are reported as the mean ± SD. ** *p* < 0.001; * *p* < 0.05.

**Table 2 ijerph-19-15835-t002:** Label-free quantitative analysis of proteins identified from *V. lateralis* muscle biopsies. Overexpressed proteins with Rsc ≥ 1.40 in the VPG vs. CG are shown.

Accession	Description	Gene	Rsc ^a^
**Q9NQX4**	Unconventional myosin-Vc	MYO5C	4.51
**Q9UNM6**	26S proteasome non-ATPase regulatory subunit 13	PSMD13	3.72
**P02774**	Vitamin D-binding protein	GC	3.72
**Q9NX40**	OCIA domain-containing protein 1	OCIAD1	3.43
**P04632**	Calpain small subunit 1	CAPNS1	3.43
**P50991**	T-complex protein 1 subunit delta	CCT4	3.43
**Q8NBN7**	Retinol dehydrogenase 13	RDH13	3.43
**Q5HYK3**	2-Methoxy-6-polyprenyl-1,4-benzoquinol methylase, mitochondrial	COQ5	3.43
**O75891**	Cytosolic 10-formyltetrahydrofolate dehydrogenase	ALDH1L1	3.43
**P11532**	Dystrophin	DMD	3.43
**O95182**	NADH dehydrogenase (ubiquinone) 1 alpha subcomplex subunit 7	NDUFA7	3.07
**Q9UFN0**	Protein NipSnap homolog 3A	NIPSNAP3A	3.07
**P55039**	Developmentally regulated GTP-binding protein 2	DRG2	3.07
**Q9NUJ1**	Mycophenolic acid acyl-glucuronide esterase, mitochondrial	ABHD10	3.07
**Q9H9P8**	L-2-hydroxyglutarate dehydrogenase, mitochondrial	L2HGDH	3.07
**O43464**	Serine protease HTRA2, mitochondrial	HTRA2	3.07
**Q15181**	Inorganic pyrophosphatase	PPA1	3.07
**O15399**	Glutamate receptor ionotropic, NMDA 2D	GRIN2D	3.07
**O60229**	Kalirin	KALRN	3.07
**Q00G26**	Perilipin-5	PLIN5	3.07
**Q13203**	Myosin-binding protein H	MYBPH	3.07
**Q8N3L3**	Beta-taxilin	TXLNB	2.58
**P49773**	Histidine triad nucleotide-binding protein 1	HINT1	2.58
**P21964**	Catechol O-methyltransferase	COMT	2.58
**P61353**	60S ribosomal protein L27	RPL27	2.58
**Q15404**	Ras suppressor protein 1	RSU1	2.58
**Q9UPY8**	Microtubule-associated protein RP/EB family member 3	MAPRE3	2.58
**Q8WUY1**	Protein THEM6	THEM6	2.58
**O96008**	Mitochondrial import receptor subunit TOM40 homolog	TOMM40	2.58
**Q9NQ50**	39S ribosomal protein L40, mitochondrial	MRPL40	2.58
**O00743**	Serine/threonine-protein phosphatase 6 catalytic subunit	PPP6C	2.58
**P15880**	40S ribosomal protein S2	RPS2	2.58
**Q5T3I0**	G patch domain-containing protein 4	GPATCH4	2.58
**Q7Z3D6**	UPF0317 protein C14orf159, mitochondrial	DGLUCY	2.58
**P54868**	Hydroxymethylglutaryl-CoA synthase, mitochondrial	HMGCS2	2.58
**Q13555**	Calcium/calmodulin-dependent protein kinase type II subunit gamma	CAMK2G	2.58
**Q969N2**	GPI transamidase component PIG-T	PIGT	2.58
**Q92523**	Carnitine O-palmitoyltransferase 1, muscle isoform	CPT1B	2.58
**Q13554**	Calcium/calmodulin-dependent protein kinase type II subunit beta	CAMK2B	2.58
**Q96AQ6**	Pre-B-cell leukemia transcription factor-interacting protein 1	PBXIP1	2.58
**P43243**	Matrin-3	MATR3	2.58
**P07384**	Calpain-1 catalytic subunit	CAPN1	2.58
**Q02641**	Voltage-dependent L-type calcium channel subunit beta-1	CACNB1	2.58
**Q9NVI1**	Fanconi anemia Group I protein	FANCI	2.58
**O14936**	Peripheral plasma membrane protein CASK	CASK	2.58
**P39059**	Collagen alpha-1 (XV) chain	COL15A1	2.58
**Q92900**	Regulator of nonsense transcripts 1	UPF1	2.58
**Q9Y490**	Talin-1	TLN1	2.58
**Q8NEZ4**	Histone-lysine N-methyltransferase 2C	KMT2C	2.58
**Q9NZJ6**	Hexaprenyldihydroxybenzoate methyltransferase, mitochondrial	COQ3	2.38
**O94874**	E3 UFM1-protein ligase 1	UFL1	2.38
**Q5T440**	Putative transferase CAF17, mitochondrial	IBA57	2.38
**Q15642**	Cdc42-interacting protein 4	TRIP10	2.38
**Q16795**	NADH dehydrogenase (ubiquinone) 1 alpha subcomplex subunit 9, mitochondrial	NDUFA9	2.20
**Q9H2U2**	Inorganic pyrophosphatase 2, mitochondrial	PPA2	2.13
**Q9H799**	Ciliogenesis and planar polarity effector 1	CPLANE1	2.13
**P02675**	Fibrinogen beta chain	FGB	2.13
**P40123**	Adenylyl cyclase-associated protein 2	CAP2	2.13
**P42765**	3-Ketoacyl-CoA thiolase, mitochondrial	ACAA2	2.10
**Q9HC38**	Glyoxalase domain-containing protein 4	GLOD4	1.85
**P61019**	Ras-related protein Rab-2A	RAB2A	1.85
**P35270**	Sepiapterin reductase	SPR	1.85
**P67809**	Nuclease-sensitive element-binding protein 1	YBX1	1.85
**Q9H6K4**	Optic atrophy 3 protein	OPA3	1.85
**Q15124**	Phosphoglucomutase-like protein 5	PGM5	1.85
**Q02224**	Centromere-associated protein E	CENPE	1.85
**P45954**	Short/branched chain-specific acyl-CoA dehydrogenase, mitochondrial	ACADSB	1.85
**Q9H0P0**	Cytosolic 5’-nucleotidase 3°	NT5C3A	1.76
**P10809**	60 kDa Heat shock protein, mitochondrial	HSPD1	1.76
**P56134**	ATP synthase subunit f, mitochondrial	ATP5J2	1.64
**O75150**	E3 ubiquitin-protein ligase BRE1B	RNF40	1.64
**Q15746**	Myosin light chain kinase, smooth muscle	MYLK	1.64
**P42704**	Leucine-rich PPR motif-containing protein, mitochondrial	LRPPRC	1.64
**P53597**	Succinyl-CoA ligase (ADP/GDP-forming) subunit alpha, mitochondrial	SUCLG1	1.57
**P62701**	40S ribosomal protein S4, X isoform	RPS4X	1.54
**Q0VFZ6**	Coiled-coil domain-containing protein 173	CCDC173	1.48
**P46778**	60S ribosomal protein L21	RPL21	1.48
**P07741**	Adenine phosphoribosyltransferase	APRT	1.48
**Q02338**	D-beta-hydroxybutyrate dehydrogenase, mitochondrial	BDH1	1.48
**O75915**	PRA1 family protein 3	ARL6IP5	1.48
**P62081**	40S ribosomal protein S7	RPS7	1.48
**Q13557**	Calcium/calmodulin-dependent protein kinase Type II subunit delta	CAMK2D	1.48
**Q9Y512**	Sorting and assembly machinery component 50 homolog	SAMM50	1.48
**Q00839**	Heterogeneous nuclear ribonucleoprotein U	HNRNPU	1.48
**P07919**	Cytochrome b-c1 complex subunit 6, mitochondrial	UQCRH	1.48
**P62899**	60S ribosomal protein L31	RPL31	1.48
**Q99733**	Nucleosome assembly protein 1-like 4	NAP1L4	1.48
**Q9UKU7**	Isobutyryl-CoA dehydrogenase, mitochondrial	ACAD8	1.48
**P15088**	Mast cell carboxypeptidase A	CPA3	1.48
**Q96A26**	Protein FAM162A	FAM162A	1.48
**Q15555**	Microtubule-associated protein RP/EB family member 2	MAPRE2	1.48
**P21817**	Ryanodine receptor 1	RYR1	1.44

^a^ Rsc represents the log_2_ ratio between the protein expression level of the muscle biopsies of VPG and those of the CG. Proteins with Rsc ≥ 1.40 or ≤ −1.40 were considered to be differentially expressed.

**Table 3 ijerph-19-15835-t003:** Label-free quantitative analysis of proteins identified from *V. lateralis* muscle biopsies. Underexpressed proteins with Rsc ≤ −1.40 in the VPG vs. CG are shown.

Accession	Description	Gene	Rsc ^a^
**P51911**	Calponin-1	CNN1	−4.78
**Q01995**	Transgelin	TAGLN	−4.60
**P02790**	Hemopexin	HPX	−3.83
**Q14847**	LIM and SH3 domain protein 1	LASP1	−3.65
**Q9UNZ2**	NSFL1 cofactor p47	NSFL1C	−3.65
**P13798**	Acylamino-acid-releasing enzyme	APEH	−3.44
**P48147**	Prolyl endopeptidase	PREP	−3.44
**P02679**	Fibrinogen gamma chain	FGG	−3.20
**P62280**	40S ribosomal protein S11	RPS11	−3.20
**Q2TBA0**	Kelch-like protein 40	KLHL40	−3.20
**Q9Y6B6**	GTP-binding protein SAR1b	SAR1B	−2.91
**Q15370**	Transcription elongation factor B polypeptide 2	TCEB2	−2.91
**Q969G5**	Protein kinase C delta-binding protein	PRKCDBP	−2.91
**P02743**	Serum amyloid P-component	APCS	−2.91
**P21980**	Protein-glutamine gamma-glutamyltransferase 2	TGM2	−2.91
**Q13526**	Peptidyl-prolyl cis-trans isomerase NIMA-interacting 1	PIN1	−2.91
**P0CW22**	40S ribosomal protein S17-like	RPS17L	−2.91
**O15145**	Actin-related protein 2/3 complex subunit 3	ARPC3	−2.91
**Q13561**	Dynactin subunit 2	DCTN2	−2.91
**P31153**	S-adenosylmethionine synthase isoform type-2	MAT2A	−2.91
**P22061**	Protein-L-isoaspartate(D-aspartate) O-methyltransferase	PCMT1	−2.84
**P49189**	4-Trimethylaminobutyraldehyde dehydrogenase	ALDH9A1	−2.68
**P56556**	NADH dehydrogenase [ubiquinone] 1 alpha subcomplex subunit 6	NDUFA6	−2.55
**O14602**	Eukaryotic translation initiation factor 1A, Y-chromosomal	EIF1AY	−2.55
**P31942**	Heterogeneous nuclear ribonucleoprotein H3	HNRNPH3	−2.55
**P30566**	Adenylosuccinate lyase	ADSL	−2.55
**Q9C0G0**	Zinc finger protein 407	ZNF407	−2.55
**Q8IUG5**	Unconventional myosin-XVIIIb	MYO18B	−2.55
**Q9GZZ1**	N-alpha-acetyltransferase 50	NAA50	−2.55
**Q00765**	Receptor expression-enhancing protein 5	REEP5	−2.55
**P62491**	Ras-related protein Rab-11A	RAB11A	−2.55
**P62195**	26S protease regulatory subunit 8	PSMC5	−2.55
**P01011**	Alpha-1-antichymotrypsin	SERPINA3	−2.55
**Q16853**	Membrane primary amine oxidase	AOC3	−2.55
**P36776**	Lon protease homolog, mitochondrial	LONP1	−2.55
**Q7L7X3**	Serine/threonine-protein kinase TAO1	TAOK1	−2.55
**P78527**	DNA-dependent protein kinase catalytic subunit	PRKDC	−2.55
**P51884**	Lumican	LUM	−2.29
**P61026**	Ras-related protein Rab-10	RAB10	−2.24
**P20774**	Mimecan	OGN	−2.19
**O94760**	N(G),N(G)-dimethylarginine dimethylaminohydrolase 1	DDAH1	−2.06
**P38646**	Stress-70 protein, mitochondrial	HSPA9	−2.06
**P20618**	Proteasome subunit beta type-1	PSMB1	−2.06
**P61254**	60S ribosomal protein L26	RPL26	−2.06
**Q15185**	Prostaglandin E synthase 3	PTGES3	−2.06
**P07203**	Glutathione peroxidase 1	GPX1	−2.06
**Q96IU4**	Alpha/beta hydrolase domain-containing protein 14B	ABHD14B	−2.06
**Q9H8H3**	Methyltransferase-like protein 7A	METTL7A	−2.06
**P55042**	GTP-binding protein RAD	RRAD	−2.06
**P50454**	Serpin H1	SERPINH1	−2.06
**Q99807**	Ubiquinone biosynthesis protein COQ7 homolog	COQ7	−2.06
**P54619**	5’-AMP-activated protein kinase subunit gamma-1	PRKAG1	−2.06
**O43765**	Small glutamine-rich tetratricopeptide repeat-containing protein alpha	SGTA	−2.06
**Q9BUB7**	Transmembrane protein 70, mitochondrial	TMEM70	−2.06
**Q9Y3B7**	39S ribosomal protein L11, mitochondrial	MRPL11	−2.06
**P27169**	Serum paraoxonase/arylesterase 1	PON1	−2.06
**Q00059**	Transcription factor A, mitochondrial	TFAM	−2.06
**Q99536**	Synaptic vesicle membrane protein VAT-1 homolog	VAT1	−2.06
**Q15366**	Poly(rC)-binding protein 2	PCBP2	−2.06
**P43686**	26S protease regulatory subunit 6B	PSMC4	−2.06
**P51888**	Prolargin	PRELP	−2.06
**Q9Y230**	RuvB-like 2	RUVBL2	−2.06
**P16930**	Fumarylacetoacetase	FAH	−2.06
**P48637**	Glutathione synthetase	GSS	−2.06
**P31948**	Stress-induced-phosphoprotein 1	STIP1	−2.06
**Q8NBS9**	Thioredoxin domain-containing protein 5	TXNDC5	−2.06
**O00148**	ATP-dependent RNA helicase DDX39A	DDX39A	−2.06
**Q9HCC0**	Methylcrotonoyl-CoA carboxylase beta chain, mitochondrial	MCCC2	−2.06
**Q9NZN4**	EH domain-containing protein 2	EHD2	−2.06
**P23246**	Splicing factor, proline- and glutamine-rich	SFPQ	−2.06
**Q9NUB1**	Acetyl-coenzyme A synthetase 2-like, mitochondrial	ACSS1	−2.06
**Q9P1V8**	Sterile alpha motif domain-containing protein 15	SAMD15	−2.06
**P00747**	Plasminogen	PLG	−2.06
**O14795**	Protein unc-13 homolog B	UNC13B	−2.06
**Q702N8**	Xin actin-binding repeat-containing protein 1	XIRP1	−2.06
**Q8IWN7**	Retinitis pigmentosa 1-like 1 protein	RP1L1	−2.06
**Q16787**	Laminin subunit alpha-3	LAMA3	−2.06
**P07585**	Decorin	DCN	−1.94
**P21291**	Cysteine and glycine-rich protein 1	CSRP1	−1.89
**P20073**	Annexin A7	ANXA7	−1.89
**P62993**	Growth factor receptor-bound protein 2	GRB2	−1.86
**P52895**	Aldo-keto reductase family 1 member C2	AKR1C2	−1.79
**O43488**	Aflatoxin B1 aldehyde reductase member 2	AKR7A2	−1.69
**Q9BXI3**	Cytosolic 5’-nucleotidase 1A	NT5C1A	−1.67
**P61106**	Ras-related protein Rab-14	RAB14	−1.61
**P62277**	40S ribosomal protein S13	RPS13	−1.61
**Q15084**	Protein disulfide-isomerase A6	PDIA6	−1.61
**P16083**	Ribosyldihydronicotinamide dehydrogenase (quinone)	NQO2	−1.61
**P47985**	Cytochrome b-c1 complex subunit Rieske, mitochondrial	UQCRFS1	−1.53
**P02545**	Prelamin-A/C	LMNA	−1.53
**P24844**	Myosin regulatory light polypeptide 9	MYL9	−1.51
**P09382**	Galectin-1	LGALS1	−1.51
**P28070**	Proteasome subunit beta type-4	PSMB4	−1.51
**P62269**	40S ribosomal protein S18	RPS18	−1.51
**P14543**	Nidogen-1	NID1	−1.51
**P13716**	Delta-aminolevulinic acid dehydratase	ALAD	−1.46

^a^ Rsc represents the log_2_ ratio between the protein expression level of muscle biopsies of the VPG and those of the CG. Proteins with Rsc ≥ 1.40 or ≤−1.40 were considered to be differentially expressed.

**Table 4 ijerph-19-15835-t004:** Biological processes according to the results of the DAVID functional enrichment analysis of over- and underexpressed proteins in the VPG vs. CG.

GO Terms	Proteins	*p*-Value
**Overexpressed proteins**		
Mitotic cell cycle *(GO: 0000278)*	MAPRE3, PPP6C, CENPE, PSMD13, CAMK2G, MAPRE2, CAMK2B, CAMK2D	0.004
Calcium ion transport *(GO: 0006816)*	CACNB1, CAMK2G, CAMK2B, CAMK2D, RYR1	0.007
Energy derivation by oxidation of organic compounds *(GO: 0015980)*	SUCLG1, UQCRH, NDUFA7, NDUFA9	0.043
**Underexpressed proteins**		
Protein complex assembly *(GO: 0006461)*	APCS, GRB2, PRKAG1, ELOB, TMEM70, TFAM, FGG, LONP1, TGM2, ADSL, UNC13B	9.22 × 10^−4^
Positive regulation of ubiquitin-protein ligase activity *(GO: 0051443)*	PSMB4, PSMC5, PSMB1, PSMC4, PIN1	8.48 × 10^−4^
Organic acid catabolic process *(GO: 0016054)*	DDAH1, MCCC2, PON1, FAH	0.029

**Table 5 ijerph-19-15835-t005:** Biological processes according to the results of the STRING analysis of overexpressed and underexpressed proteins in the VPG vs. CG.

GO Terms	Proteins	*p*-Value
**Overexpressed proteins**		
Nuclear-transcribed mRNA catabolic process *(GO: 0000184)*	RPL31, RPL21, RPL27, RPS2, UPF1, RPS7, RPS4	0.002
Regulation of calcium ion transport *(GO: 0051924)*	MYLK, CAMK2G, CAMK2B, CAMK2D	0.032
Oxidation–reduction process *(GO: 0055114)*	UQCRH, NDUFA7, NDUFA9, SUCLG1, ACAA2, ACADSB	0.02
**Underexpressed Proteins**		
Protein-containing complex subunit organization *(GO: 0043933)*	TCEB2, RPS17, LUM, MRPL11, TMEM70, PTGES3, FGG	0.0017
Post-translational protein modifications *(GO: 0043687)*	PSMB4, PSMC5, PSMB1, PSMC4, RAB11A, FGG, PDIA6, LGALS1, TCEB2	0.0061

## Data Availability

The mass spectrometry proteomics data have been deposited at the ProteomeXchange Consortium via the PRIDE [31] partner repository with the dataset identifier PXD037792.

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
