# Peer review of "Insight into the Molecular Signature of Skeletal Muscle Characterizing Lifelong Football Players"

_ijerph, 2022, doi:10.3390/ijerph192315835_

Round 1

Reviewer 1 Report

In this manuscript, the authors used V. lateralis muscle biopsies to collect the samples from lifelong football players and aged-matched untrained subjects. Then proteomic/metabolomic approaches were performed and the resulting data were analyzed by means of different bioinformatic tools. They found that lifelong football training can enhance muscle oxidative capacity in the elderly, by promoting fatty acids as preferential energetic substrates. In addition, the total polyamine content is higher in lifelong football players’ muscle. Lifelong football training can significantly influence the expression of proteins and metabolites involved in oxidative metabolism and muscle hypertrophy associated to successful aging.

This an overall well-performed study which is a part of systematic investigation of how lifelong football training is advantageous to successful aging. In addition, as it’s an “omic” study, it will expand our knowledge and reveal some potential targets. The experiments and analysis were done properly and rigorously. I listed some of my questions and suggestion below for the authors to consider.

1.     As the introduction part says, long-term recreational football training increases expression of key proteins involved in oxidative metabolism, mitochondrial biogenesis, DNA-repair, autophagy, and protein quality control. Heath-shock proteins (HSP), such as HSP 70/90 and components of the proteasome protein complex also increase owing to long-term recreational football training. But the top targets identified by proteomic approach in this study has limited overlap with the proteins list above. How to explain this?

2.     As for figure1 and figure3, some labels and words are too small to be seen and the resolution is not good enough. I can barely recognize the words when I zoom in.

3.     Did the author verified other proteins identified by proteomic analysis?

Author Response

Original Research

Insight into the skeletal muscle molecular signature characterizing lifelong football players.

Editor in Chief

Special Issue “Lifestyle-Induced and Aging-Induced Changes in Body Composition and Physical Fitness: Focus on Healthy Longevity

International Journal of Enviromental Research and Public Health.

Dear Editor,

on behalf of my co-authors, we are hereby re-submitting the revised manuscript, entitled “Insight into the skeletal muscle molecular signature characterizing lifelong football players” (ijerph-2034559) by Orrù et al. for publication as Original Research in the Special issue “Lifestyle-Induced and Aging-Induced Changes in Body Composition and Physical Fitness: Focus on Healthy Longevity” in International Journal of Environmental Research and Public Health.

We addressed all the concerns raised by the reviewers. All changes are highlighted in red in the tracked version of the manuscript and we provide, as requested, an item-by-item response to each of the comments made.  

Looking forward to hearing from you,

Yours sincerely,

Pasqualina Buono, Ph.D.

Dipartimento di Scienze Motorie e del Benessere

Università di Napoli “Parthenope”

Via Amm. Acton, 38 - 80133 Napoli-Italy

Tel.: +39 081 5474674

Original Research

Insight into the skeletal muscle molecular signature characterizing lifelong football players (ijerph-2034559)

Reviewer n.1

  1. As the introduction part says, long-term recreational football training increases expression of key proteins involved in oxidative metabolism, mitochondrial biogenesis, DNA-repair, autophagy, and protein quality control. Heath-shock proteins (HSP), such as HSP 70/90 and components of the proteasome protein complex also increase owing to long-term recreational football training. But the top targets identified by proteomic approach in this study has limited overlap with the proteins list above. How to explain this?

- We thank the reviewer for addressing a key issue related to the overlapping expression between mRNAs and proteins. Several studies have shown that there is little correlation between mRNA and protein expression data from same cells under similar conditions (Haider S et al. Curr Genomics 14: 91–110, 2013; Koussounadis A et al. Sci Rep5: 10775, 2015; Moghieb A et al. Am J Physiol Lung Cell Mol Physiol 315:L11–L24, 2018; Wang J et al. Methods Mol Biol1375: 123–136, 2016; Du Y et al. Am J Physiol Lung Cell Mol Physiol. 317:L347-L360, 2019). This lack of correlation could be due to both the different technical protocols for sample handling in transcriptomic and proteomic approaches, and the complexity of epigenetic regulation on RNAs and proteins.

Here, our transcriptomic and proteomic data, while not showing the same molecular species, point out the same cellular processes (oxidation metabolism, mitochondrial biogenesis, etc…) that contribute to the beneficial effects of exercise on health.

  1. As for figure1 and figure3, some labels and words are too small to be seen and the resolution is not good enough. I can barely recognize the words when I zoom in.

- We modified the labels and words according the reviewer’s request.

  1. Did the author verified other proteins identified by proteomic analysis?

- This study is part of a larger project that has been conducted by our research group for several years in collaboration with Prof Krustrup's team at SDU, Denmark. Among the proteins identified in this study, we also confirmed the proteasome 26S subunit non-ATPase 13 (PSMD13; Mancini et al. Frontiers in Physiol. 10: 132, 2019); in addition, other identified species have been verified (e.g. mitochondrial Hydroxymethylglutaryl-CoA synthase, HMGCS2) and are being studied for further investigation.

Reviewer 2 Report

The present MS is aimed at characterizing the proteomics and metabolomic in the Vastus lateralis muscle and the differences between lifelong football players and  sedentary elderly. The data output is rich and characterize clearly the differences between the two groups. It would be interesting, in a future research, to associate the data presented with the functional differences in the muscles from the two groups.  

I have only a few concerns about the work presented here, as listed below:

1. The Vastus lateralis is a muscle from the hind limb that presumably is largely used in football. Could the authors discuss on what extent the results could be valid also for other muscles of the same type (mainly composed by fast fibres) but less used in football? 

2. I'm not convinced by the statistical significance of the difference in BMI and body fat percentage between the two groups. However, given these differences, could the authors discuss how the results (the data output) could be related directly to these parameters, rather than to physical activity?

Minors:

1. Line 52, please define MET (Metabolic Equivalent Task)   

2. Please define Rsc at its first occurence (line 144) rather than in the legends of the tables. I have not been able to understand why the significance in protein ratio is assumed to be for -1.40<Rcs<+1.40? Could you please clarify ? 

Author Response

Original Research

Insight into the skeletal muscle molecular signature characterizing lifelong football players.

Editor in Chief

Special Issue “Lifestyle-Induced and Aging-Induced Changes in Body Composition and Physical Fitness: Focus on Healthy Longevity

International Journal of Enviromental Research and Public Health.

Dear Editor,

on behalf of my co-authors, we are hereby re-submitting the revised manuscript, entitled “Insight into the skeletal muscle molecular signature characterizing lifelong football players” (ijerph-2034559) by Orrù et al. for publication as Original Research in the Special issue “Lifestyle-Induced and Aging-Induced Changes in Body Composition and Physical Fitness: Focus on Healthy Longevity” in International Journal of Environmental Research and Public Health.

We addressed all the concerns raised by the reviewers. All changes are highlighted in red in the tracked version of the manuscript and we provide, as requested, an item-by-item response to each of the comments made.  

Looking forward to hearing from you,

Yours sincerely,

Pasqualina Buono, Ph.D.

Dipartimento di Scienze Motorie e del Benessere

Università di Napoli “Parthenope”

Via Amm. Acton, 38 - 80133 Napoli-Italy

Tel.: +39 081 5474674

Original Research

Insight into the skeletal muscle molecular signature characterizing lifelong football players (ijerph-2034559)

Reviewer 2

  1. The Vastus lateralis is a muscle from the hind limb that presumably is largely used in football. Could the authors discuss on what extent the results could be valid also for other muscles of the same type (mainly composed by fast fibres) but less used in football?

- We thank the reviewer for the interesting question that highlights the possible generalization of a result when the analysis is performed on only a part as opposed to the whole. In the case of muscle tissues, it is well known that the muscles responsible for lower limb movement are characterized by a variable composition of fast and slow fibers. All sports disciplines that involve moving the body in space through walking, running and jumping stimulate the muscles of the lower limbs with responses modulated also by the specific composition of each. Furthermore, it is known that the most effective stimulation for slow fibers comes from aerobic activities, while for fast fibers it comes from anaerobic activities; hence, the results can be generalized keeping in mind, however, that the magnitude of the response is dependent on the specific composition of the tissue being examined. In this study we evaluated the effect on muscles of a lifelong football training, which is characterized by high intensity anaerobic actions interspersed with periods of low intensity aerobic activities; so, we believe that both fibers received the proper stimulation from the training. Therefore, the data analyzed consider a response, correlated to the effects of physical activity on both the slow and the fast fibers of the V. lateralis muscle, which may also be generalized to the other muscles of lower limbs.

  1. I'm not convinced by the statistical significance of the difference in BMI and body fat percentage between the two groups. However, given these differences, could the authors discuss how the results (the data output) could be related directly to these parameters, rather than to physical activity?

- In our study we observe that the control group has higher levels of intermediate species of the fatty acid oxidation process in the muscle than in veterans (metabolomic data); this result could be due to the higher % of fat mass in the untrained subjects. On the other hand, in muscles of veterans, who are lifelong football players, the expression of enzymes responsible for the oxidative degradation of fatty acids is higher than in CG (proteomic data), indicating that the efficiency of the catabolic process is positively affected by PA. Taken together, these two independent outcomes (%fat mass and PA) help  define the same profiles: healthier for VPG, more prone to NCDs for the CG. Anyway, this study is part of a bigger project that is continuing with further investigations that will take into account the dietary habits of the newly recruited participants.

Minors:

  1. Line 52, please define MET (Metabolic Equivalent Task)

- We added the requested definition

  1. Please define Rsc at its first occurence (line 144) rather than in the legends of the tables. I have not been able to understand why the significance in protein ratio is assumed to be for -1.40<Rsc<+1.40? Could you please clarify ? 

- According to reviewer comment we have improved the Rsc definition in section 2.4 (LC-MS/MS analysis and protein identification and quantitation) of the revised version of the manuscript, as follow:

‘In order to perform a semi-quantitative comparative analysis, the protein abundances in each considered proteomes were expressed as Rsc, calculated according to the following formula:

Rsc =log2 [(n2+f)/(n1+f)]+log2 [(t1-n1+f)/(t2−n2+f)]

Specifically, Rsc is the log ratio of abundance between samples 1 (VPG) and 2 (CG); n1 and n2 are the SpCs for the given protein in sample groups 1 and 2, respectively; t1 and t2 are the total numbers of spectra over all of the proteins in the two sample groups; f is a correction factor set to 0.5 and used to eliminate discontinuity due to SpC = 0.

As previously demonstrated, we found a linear correlation between Rsc parameters and Fold change expressed as ratio of protein abundance in compared proteomes (Costanzo et al, Int. J. Mol. Sci. 2018, 19(11), 3580). The Rsc value of ±1.40 refers to a 40%-increase or decrease in relative protein abundance. We chose a cut-off value greater than ±1.30 (±30%), which corresponds to the upper and lower limits of a technical variability range.